# PA12 Surface Treatment and Its Effect on Compatibility with Nutritional Culture Medium to Maintain Cell Vitality and Proliferation

**DOI:** 10.3390/bioengineering11050442

**Published:** 2024-04-30

**Authors:** Norbert Ferencik, Maria Danko, Zuzana Nadova, Petra Kolembusova, William Steingartner

**Affiliations:** 1Department of Biomedical Engineering and Measurement, Faculty of Mechanical Engineering, Technical University of Kosice, 042 00 Kosice, Slovakia; norbert.ferencik@tuke.sk (N.F.); maria.danko@tuke.sk (M.D.); petra.kolembusova@tuke.sk (P.K.); 2Department of Biophysics, Faculty of Science, Pavol Jozef Safarik University in Kosice, 041 80 Kosice, Slovakia; zuzana.nadova@upjs.sk; 3Department of Computers and Informatics, Faculty of Electrical Engineering and Informatics, Technical University of Kosice, 042 00 Kosice, Slovakia

**Keywords:** polymer attached dyes, PA12 surface treatment biomedical, polyamide 12, 3D printing, compatibility, nutritional medium, gingival fibroblasts, mordant

## Abstract

This research investigates the suitability of printed polyamide 12 (PA12) and its dyed version to support cells in bioengineering applications. For this purpose, human gingival fibroblasts (hGF06) were cultured on PA-12 scaffolds that were 3D-printed by Multi Jet Fusion (MJF). The study examined the direct cultivation of cells on MJF-printed cell culture scaffolds and the effect of leachate of PA-12 printed by MJF on the cultured cells. The article presents research on the surface treatment of PA12 material used in 3D printing and the effect of automatic staining on cell vitality and proliferation in vitro. The study presents a unique device designed exclusively for staining prints made of the biocompatible material PA12 and demonstrates the compatibility of 3D-printed polyamide 12 parts stained in the novel device with a nutrient culture medium and cells. This novel PA12 surface treatment for biomedical purposes does not affect the compatibility with the culture medium, which is essential for cell viability and proliferation. Fluorescence microscopy revealed that mitochondrial fitness and cell survival were not affected by prolonged incubation with clear or dyed PA12 3D-printed parts.

## 1. Introduction

Three-dimensional printing technology is used to produce components of orthoses [1,2,3,4,5], prostheses, and surgical simulation systems to create anatomical disease models [5,6]. Additive manufacturing is useful in various biomedical and tissue engineering applications, regenerative medicine, and scaffolds, e.g., for bioreactors [7]. The tissue engineering triad [8] comprises cells, scaffolds [9], and growth-stimulating signals, which are considered the essential components of engineered tissues. Scaffolds, usually made of polymeric biomaterials, provide structural support for cell attachment [8,10] and further tissue development [11]. The key to successful tissue repair in tissue engineering is selecting scaffolds with controlled functional gradient architectures [12,13] and favorable biological properties while providing adequate mechanical support [8,14]. Such scaffolds provide a stable support structure, cell attachment [8], and promote cell growth and proliferation [10], ultimately leading to better tissue regeneration. The selection of appropriate scaffold shapes and materials that possess these critical properties for cell cultivation [8] is essential for the best possible tissue repair outcomes [13].

The choice of suitable material and technology and the design of a model for biomedical 3D printing must be based on specific applications [3,4,15]. For this purpose, the Multi Jet Fusion technology (HP MJF) [1,16,17], based on sintering powder, developed by Hewlett-Packard (HP) Inc. (Palo Alto, CA, USA), was used. Polymer powder, fusing, and detailing agents are used to achieve optimum performance with MJF technology [18]. Polymer additive manufacturing involves spreading polymer powder, melting it at a high rate, and crystallizing it upon cooling [1,19,20,21,22,23]. IR lamps are crucial for promoting uniform fusion of each part layer [24,25]. This is important to achieve the desired results. The process is repeated by spreading a fresh layer of powder over the first layer to build up the 3D model [2,5,11,26]. This technique is more productive and cost-effective than other methods, as it consumes less energy and material volume per unit [24]. In addition, MJF can print 30 million drops per second across the printing space’s width [18,19,27], resulting in highly accurate dimensional precision (±0.2%) [26] and correctable porosity [7]. This technique does not need chemical post-processing to attain the necessary mechanical qualities or eliminate undesirable material. Due to their high print volume, MJF printers can directly produce large parts of multi-part assemblies with moving components, eliminating the need for fasteners or other ordinary components [25]. All these advantages indicate the strong likelihood that MJF-printed products will improve mechanical performance.

One of the most commonly used materials today is medical-grade Polyamide 12 (PA12), also known as nylon 12, which is chemically inert [28,29]. It does not elicit a significant immune response [7,29] or trigger allergic reactions in most individuals. It is also considered to have low toxicity [7,30,31,32]. It does not release harmful substances or degrade into toxic by-products that could negatively affect the human body or cells. PA12 can withstand standard sterilization methods such as gamma radiation, ethylene oxide (EtO) sterilization, and autoclaving without significant degradation. Compared to other 3D printing processes for printing polyamide 12 (PA12), powder-bed fusion techniques like MJF have a distinct advantage since geometry components can be created in any orientation [16] without the need of any support material [25].

One crucial material disadvantage is the absence of microbial resistance [30]. The ability of cells to attach to a scaffold is influenced by its surface characteristics such as hydrophilicity, charge density, and chemical specificity, while its bulk chemical properties play a role in cell signaling and infiltration [33]. Both surface and bulk chemistry work together to regulate cell growth, migration, differentiation, ECM synthesis, and tissue morphogenesis, and are essential for achieving scaffold biocompatibility. Chemical modification of the scaffold has been shown to be an effective way to produce bio-chemical specificity and recognition to regulate cell behavior, direct inflammatory and immunological response, and enhance the foreign body response at the scaffold–tissue interface [8,9,34].

There have been many advances in creating highly tunable and functional scaffolds, but challenges remain before scaffold-based tissue engineering can become widely used. The scaffold’s chemical, morphological, and mechanical properties must be tuned to optimize interactions with cells and surrounding tissue, while the rate of biodegradation must be controlled to maintain the scaffold’s integrity until the growing tissue matures [33]. The scaffold should be designed with an understanding of the biochemo-mechanical properties of the native tissue and the complex mechanisms that control cellular interactions. As a result, the role of the scaffold has shifted from a passive carrier to a bioactive environment [10] with tailor-made properties for the regeneration of specific tissues [8,9,33,34].

Therefore, the study also focuses on the change of PA12 properties by a mordant medium on compatibility with the culture medium, which is essential for cell vitality and proliferation. The aim was to test the hypothesis that the staining would modify cell adhesion. The primary goal of the experiment was to determine whether the stained material in the MorPA device would affect cell viability. The motivation for this research was to modify the surface morphology and topography of the scaffold using a mordant, which could significantly alter the material’s properties such as surface roughness, porosity reduction, support of cell adhesion, hydrophilicity, or higher mechanical resistance while maintaining the matrix’s mechanical properties. One practical application of this idea is that smoothing treatment and reduced roughness could reduce tissue irritation [7] and protect against infection, which is also a reason why the research has focused on this area. Another potential application of using a mordant is to improve the ease of cleaning the surface, which could help neutralize unpleasant odors from medical devices [1]. One of the bioengineering reasons for the experiment with the staining of 3D prints from the PA12 material was based on the need for a clear visual distinction of the samples in the bioreactor and for future research.

Polymer-linked synthetic dyes are becoming increasingly popular in technical and medical applications, offering versatility and eco-friendliness [35,36]. They offer several advantages over low-molecular compounds, including reduced toxicity [35,37,38], increased recovery and reusability of materials or systems, and improved specific attributes such as surface esthetics [39]. Dyes are extensively utilized as optical sensors in medicine, chemical research, and for analytical purposes [39,40,41]. In dye-polymer chemistry, supramolecular interactions play a crucial role, and it is not mandatory to covalently attach the dye to the polymer. In addition, specific dyes’ affinity to polymers can be exploited for purification [39].

The presented research aims to provide information about our developed automatic pickling device and investigate the treated surface’s effect on cell proliferation. So far, none of the studies have yet focused on the staining of prints, which can provide aesthetic and mechanical benefits to the material. Currently, considerable attention is paid to bio-mordants for dyeing nylon [41] fiber textiles [40], which stimulated our interest in realizing the staining of 3D-printed prints from PA12. 

This research explores the suitability of MJF-printed PA-12 and its stained version as a support for cells in tissue engineering applications. For this purpose, Human Gingival Fibroblasts (hGF06) were cultured on PA-12 scaffolds that were 3D-printed with MJF. The study investigated the effect of stained PA-12 printouts on cells that are cultured directly on MJF-printed scaffolds.

## 2. Materials and Methods

The printing material used was medical-grade Polyamide 12 (HP 3D High Reusability PA 12), which is characterized by a semi-crystalline internal structure [1,24] that ensures the highest quality output. The material used in HP MJF, Polyamide 12 (PA 12), has undergone various tests and certifications for biocompatibility [7,29,42]. It has been tested for USP Class I–VI [42,43,44], which includes irritation, acute systemic toxicity, and implantation, as well as cytotoxicity (according to Cytotoxicity—ISO 10993-5, Biological evaluation of medical devices—part 5: Tests for in vitro cytotoxicity [42]) and sensitization (following Sensitization—ISO 10993-10, Biological evaluation of medical devices—Part 10: Tests for irritation and skin sensitization [42]). It also meets the requirements of USP Class I–VI and the US FDA’s guidance for Intact Skin Surface Devices. By referring to the material datasheet for this thermoplastic material [34], Powder Melting Point (DSC): 187 °C (Normative ASTM D3418 [45]); Melting Temperature: 180 °C; Density of parts: 1.01 g/cm^3^ (Normative ASTM D792 [46]); Average grain-particle size: 60 µm (Normative ASTM D3451 [47]); Bulk density: 0.425 g/cm^3^ (Normative ASTM D1895 [48]); Tensile strength: 50 mm/min, 50 MPa; Modulus of elasticity (tensile test): 1 mm/min, 1800 MPa; Modulus of elasticity (bending test): 2 mm/min, 10 N, 1700 MPa [43,44].

The scaffolds to be tested were printed with a biocompatibility certificate using a powder-based additive manufacturing 3D printer, the HP MultiJet Fusion 5200 (HP MJF) (Inc. HP, Palo Alto, CA, USA). To control the energy deposition at the voxel level, which ultimately causes the selective melting of the powder material, the MJF method uses the selective deposition of agents (such as fusing agents, detailing agents, and property-changing agents) at the pixel level. 

All research samples had the same grid design (Figure 1), presenting a primitive surface [12,13], lattice structures, and a grid with approximate dimensions of 45 × 30 mm and with a 1.5 × 1.5 mm square component—a composite scaffold with optimal gradient [12,13]. The architecture was uniform at 75% with consistent porosity. Ballotine blasting with a grain size of 0.077–0.11 mm was used as an abrasive for the final treatment of the surface of all prints. Later in the article, more detailed differences are shown between samples at magnification 200× in Figure 2. 

### 2.1. Polyamide 12 Dyeing Process

The frameworks were dyed in a device that was specially developed for automatic PA12 mordanting (MorPA [49]). The MorPA device is based on a washing machine type that is missing its original control parts. Multiple input and output processes are managed by MorPA. Temperature measurements (an NTC sensor (EPCOS, Tokyo, Japan) and an MAX6675 driver (Analog Devices, Wilmington, MA, USA) are used to monitor the temperature inside the drum); a pressure level gauge is used to measure the water level; an integrated microswitch in the door lock is used to verify the door lock’s control. The built-in electromagnetic solenoids control the water injection and stop the pickling medium from filling up; the H-bridge L298N (STMicroelectronics, Geneva, Switzerland) controls the motor rotation speed; the door can be locked and unlocked; the pickling drum rotates due to a DC motor of the PGS430 series (Gros Company, Brno, Czech Republic) with a planetary gearbox and a 294:1 ratio, which has a rated torque of 25 kg. cm and a rated speed of 18.9 rpm at 12 V [49].

Mordanting and dyeing require two primary cycles. The first stage of the mordanting cycle has the following sequence: (1) adding the pickling medium to the water, (2) bringing the water to the correct pickling temperature, (3) pickling by adjusting the angle of the drum swing, (4) heating of the pickling medium continuously, and (5) removing the pickling medium and turning off the heat source [49].

Once the dyeing cycle is completed, the pieces undergo a stabilization process divided into multiple equal steps: (1) filling the drum with clear water, (2) first washing the parts with clean cold water, (3) heating the cold water to the necessary temperature, (4) tilting the drum to the necessary angle for stability, and (5) emptying the stabilization water [39,49].

During the mordanting process, the parts are immersed in an acidic solution containing color and additives [39,49,50]. A freely available mordant (N-RIT-88150-BUN1 (Nakoma Products LLC, Bridgeview, IL, USA)) was used but this was not tested on animals. According to the 2012 OSHA Hazard Communication Standard [51], this product does not contain any hazardous substances. Based on the supplied information, the product does not present an acute toxicity [51,52] hazard. This product is not suitable for human use and was used only for experimental purposes [52]. The procedure for dyeing polyamide involves mixing the acid dye with hot water and a mild acid [50] such as vinegar or citric acid. The samples are added after heating the dye solution to around 82 °C. This is followed by mixing the material in the dye solution for 30 min to 1 h, depending on the desired color intensity. The exact time depends on the specific dye used and the desired color. The dyed 3D-print sample is stabilized through a cold shower, which washes away unbound paint residues. This is followed by exposing samples to the influence of a warm water bath with a mild detergent. A repeated cycle with cold-water action removes the remaining unbound dye and improves color stability and subsequent free air-drying. 

Each stained specimen in N-RIT-88150-BUN1 mordant had a process setting of 85 pickling cycles at 88 degrees Celsius and 25 stabilization cycles at 55 degrees Celsius [49].

#### 2.1.1. Preparation of the Sample 

A stock solution (10 mg/mL) of MorPA was prepared. The black powder of MorPA was dissolved in sterile PBS. Appropriate volumes of the stock solution were added to the 1 mL of complete culture medium to achieve dye concentrations 0.0025 mg/mL, 0.005 mg/mL, 0.01 mg/mL, 0.04 mg/mL, 0.1 mg/mL, 0.25 mg/mL, 0.5 mg/mL, 1.0 mg/mL, 2.5 mg/mL, and 5 mg/mL in culture medium. In the second procedure, the clear PA12 and dyed PA12 printed parts were added directly to the complete the culture medium and were incubated for 48 h. A 0.22 µm filter was used to remove possible impurities and the medium was added to the seeded cells.

#### 2.1.2. UV-Vis Absorption

Measurements of UV-Vis absorption was conducted in a UV-2401PC Shimadzu spectrophotometer (Shimadzu Corporation, Kyoto, Japan) with a slit width corresponding to a resolution of 1.0 nm in the 250–700 nm range. The Origin Pro Program 2021 analyzed the obtained data.

#### 2.1.3. Cell Culture [53]

Human gingival fibroblasts (hGF06) [54] were used in all experiments. The cells were regularly kept in Dulbecco’s modified Eagle’s medium (DMEM) with high glucose content. This medium was supplemented with 10% fetal bovine serum (FBS), penicillin (50 μg mL^−1^), and streptomycin (50 μg mL^−1^). The cells were kept in a 5% CO_2_ humidified atmosphere at 37 °C. All the chemicals used in the experiment were Sigma-Aldrich products (Darnstadt, Germany).

#### 2.1.4. Experimental Cell Culture Conditions and Controls [55]

Complete cultivation media containing dissolved dye powder (the final concentrations of dye in the medium were 0.005 mg/mL, 0.01 mg/mL, 0.04 mg/mL, 0.1 mg/mL, 0.25 mg/mL, 0.5 mg/mL, 1.0 mg/mL, 2.5 mg/mL, and 5 mg/mL) were prepared and used for cultivation with seeded cells. In the second method, clear PA12 and dyed PA12 printed parts were placed into a complete cultivation medium and left to incubated for 48 h at 37 °C [56,57]. After this, the medium was then filtered and used for 24, 48, and 72 h of cultivation with seeded cells [53,54]. Non-affected cells and cells incubated with the photoactive compound hypericin (4,5,7,4′,5′,7′-Hexahydroxy-2,2′-dimethylnaphthodianthrone dissolved in DMSO, stock solution: 2 mg/mL), which induces cell death after irradiation, were used as controls.

#### 2.1.5. Hypericin Photoactivation

Cells were seeded in a 96-well plate and incubated overnight. An amount of 5 × 10^−5^ M hypericin was added and the cells were incubated in dark conditions for 1 h at 37 °C. After that, the medium was replaced with a fresh one without hypericin, and a mono-chromatic homemade diode illuminator was used to irradiate the cells at a wavelength of 590 nm and a light dose of 2 J/cm^2^. The cellular response was observed 24, 48, and 72 h after irradiation [58,59].

#### 2.1.6. Cell Viability Assay

By using a colorimetric technique based on metabolically active cells reducing a yellow tetrazolium salt (3-(4,5-dimethylthiazol-2-yl)-2,5-diphenyltetrazolium bromide) to purple formazan crystals, cell viability was assessed. Cells (10^4^ cells per well) were seeded in a 96-well plate and incubated overnight. Clear PA12, dyed PA12 printed parts, and medium containing 0.005 mg/mL, 0.01 mg/mL, 0.04 mg/mL, 0.1 mg/mL, 0.25 mg/mL, 0.5 mg/mL, 1.0 mg/mL, 2.5 mg/mL, and 5 mg/mL of MorPA dye was added and the cells were incubated 24, 48, and 72 h, respectively, 120 and 168 h (for clear and dyed PA12 printed parts). After the indicated time intervals, the medium was removed and replaced with fresh medium containing 10 µL MTT (Sigma-Aldrich, stock solution 5 mg/mL). Samples were incubated for four hours at 37 °C [53,54]. After incubation, the medium was removed, and the formazan was dissolved in dimethyl sulfoxide (DMSO). The absorbance was measured at 570 nm using a microplate reader (GloMax™-Multi+Detection System (Madison, WI, USA) with Instinct Software (https://www.instinct.vet/, accessed on 7 March 2024) (Doylestown, PA, USA). Annexin V-FITC (BioLegend, San Diego, CA, USA) and propidium iodide (PI) (Sigma-Aldrich, Darnstadt, Germany) were used to detect apoptotic and dead cells. Cells (10^6^ cells per sample) were washed with cold PBS (Sigma-Aldrich, Darnstadt, Germany) and resuspended in 100 µL Annexin-V binding buffer (BioLegend, San Diego, CA, USA). An amount of 5 µL of Annexin V-FITC was added, and samples were incubated for 15 min. in the dark at room temperature. The sample volume was made up to 0.5 mL. Just before the measurement, 10 µL of PI was added to each sample and fluorescence of FITC and PI was measured [55,60].

#### 2.1.7. Confocal Fluorescence Microscopy

2 × 10^5^ cells were transferred to a 3.5 mm glass-bottomed Petri dish and incubated overnight. Clear PA12 and dyeing PA12 material were incubated with the cells for 7 days. Cells were washed with DMEM (Sigma-Aldrich, Darnstadt, Germany) and incubated with Hoechst33342 (Sigma-Aldrich, Darnstadt, Germany) and 200 nM MitoTracker^®^Orange CMTM/ROS (ThermoFisher Scientific, Waltham, MA, USA) for 30 min at 37 °C. The staining solution was prepared in 1 mL of culture media. After staining, the samples were washed three times, and fluorescence was detected by using a confocal microscope LSM700 Axiovert (Zeiss, Oberkochen, Germany) at 405 and 555 nm excitation. The fluorescence signal was analyzed using ZEN 33 software (Zeiss, Oberkochen, Germany).

Images of non-labeled and labeled material were analyzed by digital light microscope SONY IMX290 (Lapsun, Shenzen, China) with 200×–2000× Inspection Zoom Monocular C-mount Lens and visualized with the built-in SONY camera software (Version 1.2). The images were then processed in the ImageJ-FIJI software (Version 2.9.0.) (National Institutes of Health, Bethesda, MD, USA) to quantify and characterize the differences between samples at magnification 200× (Figure 2). It developed an analysis workflow to compare the gap sizes between the grains of the material before and after labeling. After the precision set-up of the threshold, the next step in the image processing was to use the “Analyze particles” tools in ImageJ and calculate the area of the gaps.

## 3. Results

The culture media used for in vitro cell cultivation contain the essential components needed to maintain cell viability and proliferation of cells. Depending on the cell type, the culture medium contains a defined mix of salts, vitamins, carbohydrates, amino acids, fatty acids, buffers, and supplements such as serum, which is the source of growth factors, hormones, and adhesion factors [55,60,61]. At the same time, the culture medium provides an environment with a guaranteed osmolality level and pH that can be visually controlled through phenol red, a pH indicator that is usually used at a concentration of 5–10 mg/mL. Figure 3a shows the absorption spectrum of DMEM supplemented with 10% FBS and 1% antibiotic in the presence of dye powder in the spectral range of 200–700 nm. In the region of 200–300 nm, two absorption bands are evident with maxima at 240 nm and 275 nm; both are related to the presence of amino acids in the medium. 

In the region above 300 nm, there are two distinct absorption bands related to the presence of Phenol red. The first has a maximum at 560 nm, while the second is less prominent and has a maximum at 415 nm. Despite an increase in the concentration of dye powder in the culture medium, there is no change in the shape of the absorption spectrum or the position of the absorption bands. Nevertheless, the medium’s absorbance increases as the concentration of the dye powder increases. The ratio of the measured values at 415/560 nm remains consistent in all tested samples, except for the last two (Figure 3b). At a concentration of 2.5–5.0 mg/mL, there is a more pronounced increase in absorbance, indicating a change in pH of the medium due to the high concentration of dissolved dye.

Figure 3 shows the absorption spectra of the complete culture medium, incubated for 12 h in the presence of clear and dyed PA12 material. The presence of clear PA12 granules does not change the shape or intensity of the absorption spectrum of the medium compared to the control. The presence of dyed PA12 material in the medium led to a slight increase in the absorbance intensity of the medium, but the shape of the spectrum remained unchanged. Based on the measured spectra and their intensities (Figure 3a), it can be concluded that less than 0.01 mg/mL of dye is released from dyed PA12 granules into the medium during the 24 h of incubation. Comparing the absorption bands for phenol red, the ratio of absorption at 415/560 nm remains the same for clear and dyed PA12 material compared to the control. Thus, the results indicate that low concentrations of dye released from PA12 material into the culture medium do not significantly alter the pH of the medium in which the cells are maintained.

Absorption at 570 nm is important for determining cell metabolic activity and survival. The absorbance at 570 nm of medium incubated with clear PA12 material was similar to the control. There was a slight increase in absorbance when dyed PA12 granules were used. This increase is related to the release of the dye from PA12 into the culture medium (Figure 4).

The MTT viability assay is a method of measuring metabolic activity and cell survival. This is achieved by measuring the absorbance of formazan at 570 nm (Figure 5a). However, an increase in absorption due to the released dye must be considered and separated out as a background. When cells are incubated in the presence of dissolved dye powder, the metabolic activity and survival of the cells are slightly decreased with increasing concentration of black powder dye in the medium. At a dye concentration of 5 mg/mL, the metabolic activity and cell survival is significantly reduced (Figure 5b). On the other hand, when cells were incubated in a medium containing clear and dyed PA12 samples, metabolic activity and cell survival were comparable to controls in all measured times including when incubation time was extended to 120 and 168 h (Figure 5c and Figure 6). These results were confirmed by measuring the fluorescence of annexin V-FITC and propidium iodide. Annexin V is a powerful tool for detecting apoptotic cells. It specifically binds to phosphatidylserine, which is expressed on the outer leaflet of the plasma membrane of apoptotic cells [57,61]. Propidium iodide passes through the membranes into the nucleus of dead cells, where it binds to DNA. The fluorescence intensities of both markers were comparable to the unaffected control, but there were slight differences in the statistical error. Mitochondria are the central organelles of cell metabolism. Their main task is to provide energy for all cellular processes and functions. As a result of the work of the electron transport chain, a mitochondrial potential (∆ψm) is created on the mitochondrial membrane. The ∆ψm fluctuations affect the functionality of mitochondrial compartment. Dissipated ∆ψm represents the point of no return, when the cell can no longer reverse the direction towards cell death. Fluorescence microscopy is a standard method for imaging of mitochondria in living cells as well as for basic evaluation of their morphology, functionality, and reactive oxygen species (ROS) generation. MitoTracker^®^Orange CMTM/ROS is a cell-permeable dye selective for mitochondria that reaches mitochondria in response to ∆ψm. The intensity of MitoTracker^®^Orange CMTM/ROS fluorescence is dependent on the presence of ∆ψm and decreases significantly if it is lost. At the same time, MitoTracker^®^Orange CMTM/ROS is considered a marker suitable for the assessment of ROS [58]. Fluorescence microscopy (Figure 6) showed that cells incubated for 168 h with clear or dyed PA12 printed parts had the same morphology of nuclei and mitochondria as control cells. Both the fluorescence intensity and MitoTracker^®^Orange CMTM/ROS localization are within the normal range, similar to control cells, which means that the mitochondrial potential on the membrane is created and the degree of oxidative stress is similar to control cells. In summary, it can be concluded that prolonged incubation time does not affect mitochondrial fitness and cell survival in the presence of clear or dyeing PA12 printed parts.

## 4. Discussion

During mechanical testing, both PA12 specimens without staining (Group 1 (G1), n = 30) and with staining (Group 2 (G2), n = 30) were subjected to tensile stress. The study evaluated the tensile properties of plastic using standardized ASTM methods. A Universal Testing Machine Inspect Table Blue (Hegewald Peschke Meß- und Prutechnic GmbH, Germany) was used. For this purpose, “dog-bone” specimens with a thickness and width of 14 mm were used. The findings provide insight into the material’s mechanical behavior under stress (tensile test), which may interest bioengineers and industry professionals. The breaking force (Fm [N]; G1: Fm = 2035 N ≈ G2: Fm = 2005 N), elasticity modulus (E [MPa]; G1: E = 590 Mpa ≠ G2: E = 1402 Mpa), and relative elongation (ε [mm]; G1: ε = 25 mm ≠ G2: ε = 41 mm) were measured. Previous studies extensively researched the chemical, physical (e.g., water tightness behavior [26]), and mechanical properties (e.g., crystallization shrinkage [19], quasi-static compressive behavior [62]) of PA12 parts printed using Multi Jet Fusion (HP MJF) [16,17]. It was found that the crystallization behavior of PA12 is associated with the cooling rates applied to the material. As per a recent study, crystallization shrinkage is linked to lamellar folding, which is affected by cooling rate. Minimizing cooling rate variation after printing is essential to achieve an excellent geometrical shape in 3D printing. Slow cooling reduces shrinkage and crystal core size, improving mechanical properties. Polymer cooling rate significantly affects crystallization, so it is crucial in additive manufacturing [19,63]. Due to the limitations in thermal stability and mechanical strength of polymer powders as compared to metal, extensive research has been conducted and is still ongoing on polymer composite powders [64,65]. These composite powders are designed to fabricate functional parts with precisely tailored mechanical properties [24,66].

In a recent study [5], the properties of 3D-printed PA12 material were measured for use as tissue equivalent. The materials were found to have a high elastic modulus and can simulate bone tissue. As the tube voltage increased, the X-ray attenuation coefficient decreased. Analysis of the CT number revealed that it was closer to adipose tissue. Despite the gap in acoustic properties with natural tissue, the material had sound speeds similar to human skin [5].

Researchers have used MJF technology to explore the potential of creating biocompatible 3D-printed PA-12 bioreactors [7,29]. In the medical field, guided bone regeneration is becoming more popular due to the irregular porous nature of human bone, which is more conducive to bone tissue formation [13]. It has been proven that Triply Periodic Minimal Surface (TPMS) [12] can overcome the limitations of regular porous scaffolds. The advantages of TPMS include variable management of the porosity gradient through parametric modeling and less stress concentration because of the zero-mean curvature [12]. Hierarchical pores in the TPMS architecture are crucial for improving fluid permeability [12]. However, Maxwell’s stability criterion does not explain the significant dependence of cell mechanical performance on loading direction [62,67]. Lattice structures, on the other hand, have a higher potential for development and effectively promote cell adhesion and migration [13] while reducing weight. The scaffold has a high potential for tissue engineering and regenerative medicine thanks to its ability to achieve higher mechanical properties [62,67]. The hydrophilicity of the scaffold is critical for cell growth and adhesion [13,34]. It is imperative to consider this factor to achieve the desired outcomes in tissue engineering and regenerative medicine. However, the manufacturing process generates porosity, which weakens the tightness requirement under fluid pressure. Coatings or infiltrations are currently needed to seal the material to ensure no leakage through porosity [26].

Although MJF technology can produce high resolution, the printed surface may still need to be rough and uneven, making it uncertain how this surface topography will impact cell adhesion, morphology, and other anchorage-dependent cellular processes [7]. PA-12’s functionalization ease also enables the enhancement or manipulation of cell adhesion [7,30,68].

Research was conducted to determine whether Multi Jet Fusion (MJF)-printed PA-12 may promote osteogenesis and cell proliferation. The findings show that MJF-printed PA-12 does not inhibit L929 fibroblast and MC3T3e1 osteoblast development. Both cell types can connect to and develop on the substrate, though not as much as on traditional polystyrene culture plates. Collagen coatings, poly-D-lysine, and plasma therapy did not considerably alter the outcomes, indicating the possibility of additional mechanisms at work. Although MJF-printed PA-12 can also promote MC3T3e1 osteogenesis, it shows distinct capacities to support the proliferation of various cell types, especially in later passages. The growth of MC3T3e1 is somewhat hampered, whereas the growth of L929 is constant from passage to passage. In addition, the study found that L929 sub-cultured back to the polystyrene plate can grow just as well as those on the standard plate, suggesting that the proliferation of cells on MJF-printed PA-12 is not irreversibly impaired. According to the results of the study, MJF-printed PA-12 may find applications in biology [7].

Surface modification techniques are used to customize the surface properties of 3D-printed material [1] to meet specific requirements, such as esthetics, the comfort of medical devices [69,70], physical and mechanical properties [31,66,70], and improving cell adhesion. However, only a limited number of studies have been conducted on surface treatments of 3D-printed PA12 products. These studies mainly focus on chemical functionalization, which involves treating the PA12 surface with chemical agents to functionalize with biomolecules or peptides that facilitate cell adhesion. For instance, coating the surface with extracellular matrix (ECM) proteins or peptides has been shown to promote cell attachment and spreading [7].

Composite materials are used to improve properties that are lacking in a single material [13]. Glass additions are often used as thermoplastic fillers because they improve load-bearing performance in the elastic deformation range. Their spherical shape allows for large packing fractions, improved flow properties, increased part isotropy, and less warpage making them a popular choice for reinforcement. Glass bead components filled with thermoplastic provide a well-defined mechanical performance for SLS [24]. In a recent study, the effect of hydroxyapatite (HA) on the polyamide 12 (PA12) scaffold’s dependability for bone grafts was investigated. The study found that a moderate amount of HA improved the scaffold’s mechanical properties, while an excessive amount of HA degraded them. Compared to the pure PA12 scaffold, the 96% PA12/4% HA scaffold exhibited a greater yield strength and compressive modulus. According to the findings, PA12/HA composites are an excellent tool in bone tissue engineering due to their strong mechanical properties and biocompatibility [2,26].

There are several practical applications of polymers in medicine that involve using dyes attached to the polymer [2,26]. These dyes can be polymerized and cross-linked to create blanks. Dyeing changes the shade and physical properties of the original parts, resulting in a new set of components. To assess the quality of the dyed product, the color of the new components can be compared with the original ones. Assuming that the product is of superior quality, adequately dried in a controlled environment, and well-pigmented and stained, the dye should stop releasing and become resistant to rubbing against fabric and paper [49]. Methacrylate anthraquinone dyes are a suitable type of monomer that can be used for a variety of purposes. These dyes can be synthesized in blue, green, and red colors and can be combined with 2-hydroxyethyl methacrylate and tetrahydrofurfuryl methacrylate. Additionally, they can be cross-linked with ethylene glycol dimethacrylate. By mixing different anthraquinone monomers, almost any color can be created. Some interesting dye categories include triphenylmethane, azo, anthraquinone, perylene, and indigoid dyes [39,71]. The process of creating dye–polymer conjugates can involve two types of binding modes—covalent and non-covalent. Covalent binding requires the formation of bonds, whereas non-covalent chemical binding can occur through various types of interactions, such as ionic and dipole–dipole molecular interactions. Non-covalent chemical binding can also transpire by forming inclusion complexes [39].

Dyes for scaffolds should be biocompatible [72], non-toxic [27], adhere to the scaffold material, be resistant to fading from light exposure, achieve the desired hue, and have minimal environmental impact [27,55]. They should interact with scaffold material and other chemicals used in the staining process [1]. They should have sufficient mechanical strength and structure to allow cells to settle and proliferate naturally, control their degradation and dissolution to avoid toxicity, and be harmless to cells. The degree of biocompatibility is influenced by the chemical composition, structure, surface properties, and degradability of the material used [10,33].

One study was conducted to test modifications on PA12 parts [1] printed using MJF processes. The study focused on modifying the surfaces of the printed samples through mechanical treatment, chemical treatment, antibacterial coatings, and dyeing. However, the research found that the type of chemical treatment and dyeing had no significant effect on the topography parameter values [1]. The aim of this research was to develop medical device manufacturing technology by investigating the effect of smoothing post-processing on the surface topography of printed elements. Applying mordant can improve the surface’s ease of cleaning and neutralize unpleasant odors. Smoothing treatment and reduced roughness can also reduce skin irritation, thereby protecting the skin from infections [1].

In recent years, there has been an increase in the use of bio-dyes, bio-mordants, pigments, and clays for creating colored materials for design purposes. Bio-based natural materials are biocompatible, safe, and do not release toxic compounds during biodegradation [72]. There is growing interest in utilizing natural sources such as herbs, fruits, vegetables, and waste products, for extracting dyes and pigments due to ecological concerns [55], cytotoxicity [55], or biocompatibility. This environmentally friendly and non-toxic approach [27] is an excellent alternative to chemical-based products [27]. Their disadvantage is the complexity of achieving color fastness, mixing, and following specific procedures based on the material for dyeing. We attempted to dye PA12 material using pomegranate seeds and walnut shells, but the most successful results were achieved using black beans. However, the bio-staining protocol must be revised to achieve the proper color intensity and ensure decreased biodegradation. 

There is a lack of information on how different post-processing treatments affect the surface roughness of 3D-printed polymeric parts. Furthermore, there are currently no established standards for treating the surface of such parts in the medical industry. Therefore, this proposed research is crucial to creating measurable parameters for the post-processing treatment of 3D-printed polymeric parts in medical applications. The present regulatory guidelines for medical product approval are ISO 13485:2016 [1,73] and SIST EN ISO 10993-1:2021 [1,74].

Most research in 3D printing has focused on creating scaffolds that have well-defined architectures and are highly customizable [7], to prepare complex personalized scaffolds [75]. In general, printing scaffolds with different morphology is a simple and universal approach used in tissue engineering. By varying the shapes and pattern of the scaffolds, it is possible to mimic the complexity of native tissues, which promotes better cell attachment, proliferation, and differentiation [9]. Various shapes of scaffolds can be used, including simple geometric shapes like cubes and cylinders, and also more complex shapes like meshes, grooved structures, or fractal patterns [9]. It is also possible to utilize various patterns on scaffolds, such as horizontal and vertical stripes or grooves, circular or elliptical holes, and more complex patterns like checkerboards, hexagons, or fractals. These patterns can have an impact on cell behavior and their interaction with the scaffold [9,67].

Several scaffold characteristics impact cell morphology, proliferation, spreading rates, and differentiation. These include architecture, fiber diameter and alignment [75], pore size and shape, microgroove patterns, and surface roughness [33]. In general, scaffolds with high porosity and large pores tend to promote increased proliferation, differentiation, and gene expression. Pores larger than a cell can cause cells to align along a single filament, while pores smaller than a cell can lead to cell bridging across filaments, resulting in slower movement and longer migration distances. Cell behavior is dependent on the type of cell and scaffold material [72]. Changes to fiber alignment or morphology can achieve different cellular responses. However, further research is needed to confirm correlations between cell shape and fiber orientation [33,75].

However, the biocompatibility of these 3D-printed scaffolds is crucial for the proper functioning of cells. In some cases, the material used for printing may not be suitable for cell growth, and research has shown that solidified polymers may contain toxic residues that inhibit cellular growth. Some researchers have tried to reduce the negative effects by employing post-printing treatments, such as exposing the material to ultraviolet light. It is also important to note that some 3D-printed medical devices implanted in patients have caused infections and allergic reactions. Therefore, it is essential to examine the biocompatibility of 3D-printed materials to minimize the risk of failure [7] in their performance. 

It is worth noting that dyeing PA 12 can be more challenging than dyeing other types of nylon due to its unique chemical composition. Therefore, it is recommended to perform a small test before dyeing larger quantities of material to ensure the desired results. In addition, it may be necessary to adjust the dyeing time and temperature based on the specific type of PA 12 used and the dye used. In our case, we used the MorPa12 device for coloring the material [49].

## 5. Conclusions

During an experiment, the clear and dyed PA12 3D-printed scaffolds were incubated in cell cultivation media that contained essential components for cell growth. It was observed that the clear PA12 granules did not change, whereas the dyed PA12 material caused a slight increase in the absorption spectrum. An MTT viability assay was conducted, which showed that metabolic activity or cell survival did not significantly reduce in the presence of clear or dyed PA12 printed parts, even after extended incubation times. Furthermore, fluorescence microscopy revealed that mitochondrial fitness and cell survival were not affected by prolonged incubation with clear or dyed PA12 printed parts.

## Figures and Tables

**Figure 1 bioengineering-11-00442-f001:**
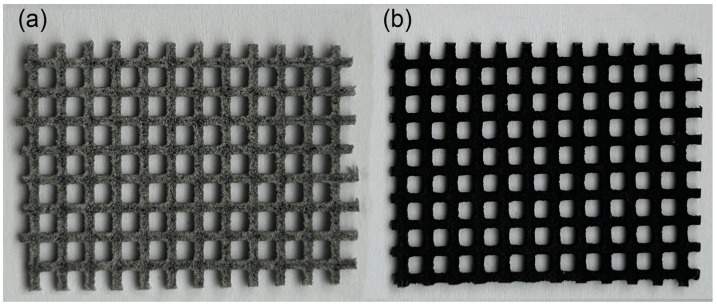
(**a**) A 3D-printout sample from PA12 (scaffold grid with the inner dimension of the square a = 3 mm and a wall thickness of 1 mm); (**b**) Stained 3D-printed sample with a duration of pickling of one hour and stabilization of 35 min. To dye the specimens, 10 g of mordant was utilized.

**Figure 2 bioengineering-11-00442-f002:**
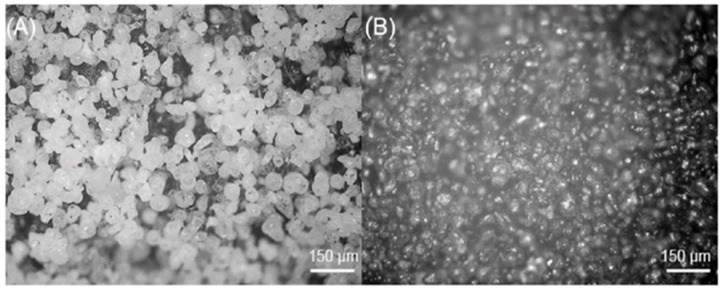
**Microscopic** Images of (**A**) 3D-printout sample from PA12 (**B**) and dyeing sample (scale bar: 150 μm) [49].

**Figure 3 bioengineering-11-00442-f003:**
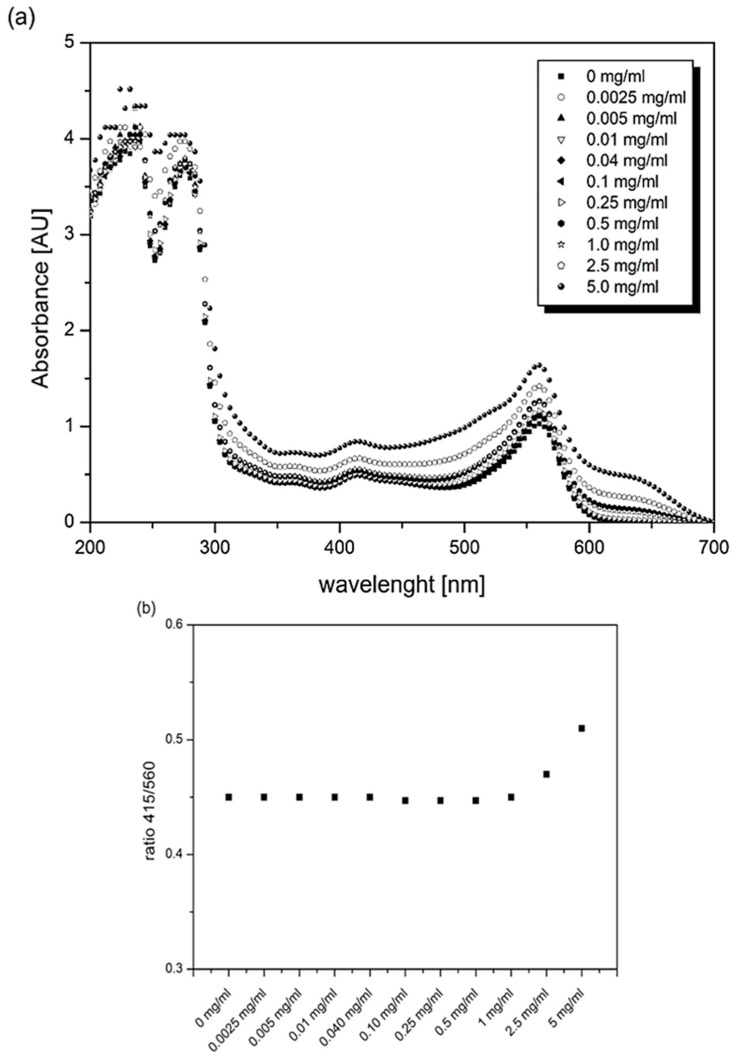
(**a**) UV-Vis absorption spectrum of DMEM supplemented with 10% FBS and 1% antibiotic in the presence of dissolved dye powder in the spectral range of 200–700 nm. (**b**) The ratio of the measured absorbances at 415/560 nm.

**Figure 4 bioengineering-11-00442-f004:**
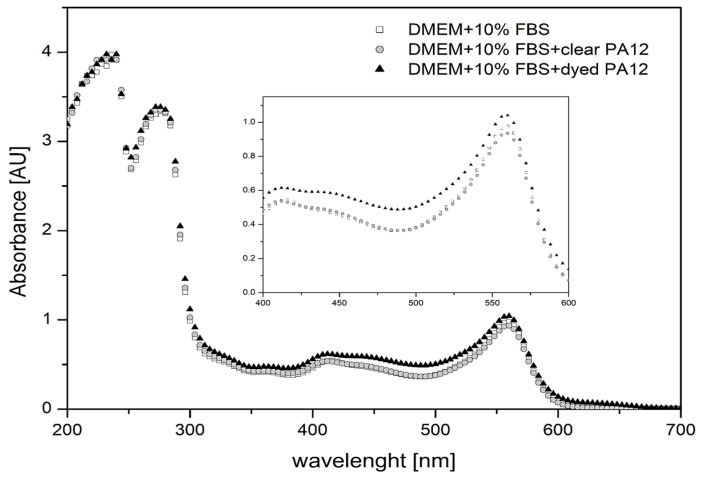
UV-Vis absorption spectra of the complete culture medium, incubated for 12 h in the presence of clear and dyed PA12 granules. Insert: Detail of absorption in the spectral range of 400–600 nm.

**Figure 5 bioengineering-11-00442-f005:**
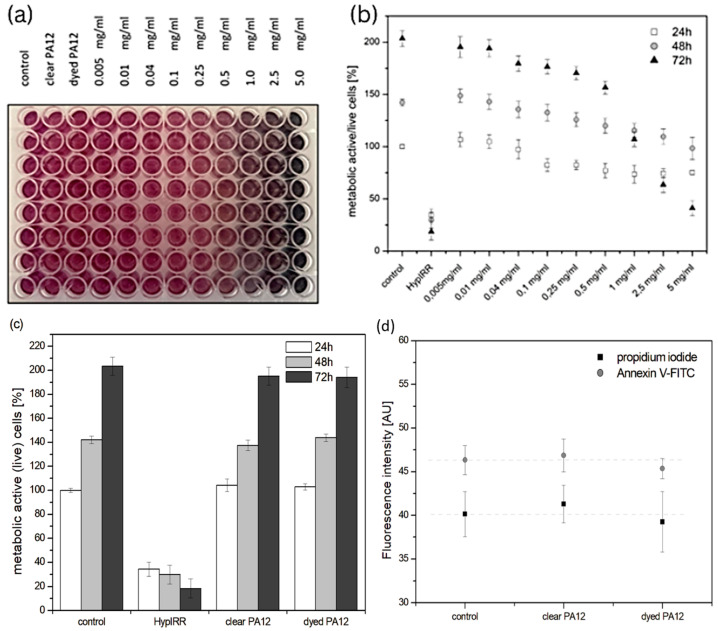
Cell viability. (**a**) MTT test preparation: 1—control, 2—clear PA12, 3—dyed PA12, 4–12—dissolved dye powder (0.005–5.0 mg/mL) (see Materials and Methods). (**b**) Metabolic activity of mitochondria (cell viability) in the presence of dye powder dissolved in culture medium. (**c**) Metabolic activity of mitochondria (cell viability) in the presence of clear PA12 and dyed PA12 printed parts. Non-affected and hypericin-loaded irradiated cells were used as controls. (**d**) Fluorescence intensities of propidium iodide and annexin V-FITC.

**Figure 6 bioengineering-11-00442-f006:**
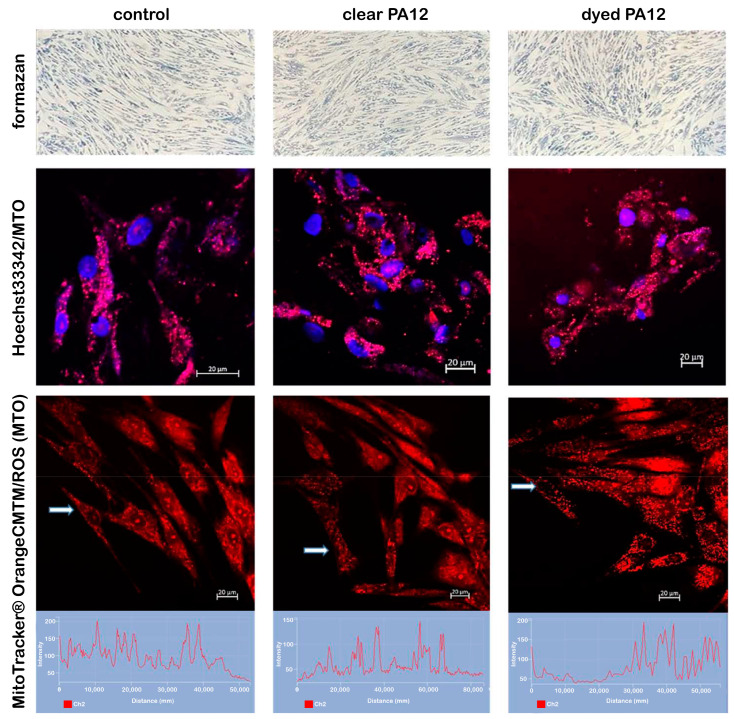
Cell viability. Microscopic images of viable fibroblasts incubated in the presence of clear and dyed PA12 printed parts: brightfield microscopy of formazan metabolizing fibroblasts, fluorescence confocal images of nuclei (blue) and mitochondria (pink), fluorescence confocal images of mitochondria (red), and representative fluorescence intensities of MitoTracker^®^Orange CMTM/ROS.

## Data Availability

Data will be provided if requested.

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
