# Peer review of "PA12 Surface Treatment and Its Effect on Compatibility with Nutritional Culture Medium to Maintain Cell Vitality and Proliferation"

_bioengineering, 2024, doi:10.3390/bioengineering11050442_

Round 1

Reviewer 1 Report

Comments and Suggestions for Authors

แอ

Reviewer’s comments:

1.       The authors mentioned in the Materials and Methods section about testing such as cytotoxicity, irritation and skin sensitization, biocompatibility, structure, cell viability assay, and mechanical testing. However, there are no results of mechanical properties to compare PA12 and PA12/HA, including compatibility results.

2.       How safe is it for use in biomedical applications? And what is its lifespan in the body?

3.       What are the magnifications of Figures 1 and 2?

Author Response

Dear Reviewer,

We greatly appreciate your devoted time to our manuscript. Thank you for your valuable comments on the research. We carefully checked grammar mistakes and other errors in the paper. We edited the study design to represent our research better and tried to add more details to make the presentation of the results more transparent.

  1. The authors mentioned in the Materials and Methods section about testing such as cytotoxicity, irritation and skin sensitization, biocompatibility, structure, cell viability assay, and mechanical testing. However, there are no results of mechanical properties to compare PA12 and PA12/HA, including compatibility results.

We added more information In: 2. Materials and Methodsabout experimental details, e.g. cell culture plate, concentrations, Confocal fluorescence microscopy, etc.

In: “3. Results”

The results of the fluorescence microscopy indicate that there was no significant difference in terms of cellular morphology, mitochondrial potential, and oxidative stress between the control cells and those exposed to clear or dyeing PA12 printed parts for 168 hours prolonged incubation at prolonged incubation with such parts did not have any negative impact on mitochondrial fitness or cell survival.

We supplemented our study by evaluating the tensile properties of plastic using standardized ASTM methods. We employed "dogbone" specimens measuring 14 millimeters in thickness and width to accomplish this. Our findings provide insight into the material's mechanical behavior under stress, which may interest academic and industry professionals alike.

In: “4. Discussion

During mechanical testing, both non-dyed (Group 1 (G1), n=30) and dyed (Group 2 (G2), n=30) PA12 samples were subjected to tensile stress. The study evaluated the tensile properties of plastic using standardized ASTM methods. It was used a Universal Testing Machine Inspect Table Blue (Hegewald Peschke Meß- und Prutechnic GmbH, Germany). To accomplish this, it employed "dogbone" specimens measuring 14 millimeters in thickness and width. The findings provide insight into the material's mechanical behavior under stress (tensile test), which may interest bioengineers and industry professionals. The breaking force (Fm [N]; G1: Fm= 2035N » G2: Fm= 2005N), elasticity modulus (E [MPa]; G1: E= 590Mpa ¹ G2: E= 1402Mpa), and relative elongation (e [mm]; G1: e = 25mm ¹ G2: e = 41mm) were measured.”

  1. How safe is it for use in biomedical applications? And what is its lifespan in the body?

These products are unsuitable for human use and were used only for experimental purposes. They are safe from the point of view of in vitro bioengineering applications.

In: Introduction:

The primary goal of the experiment was to determine whether the stained material in the MorPA device would affect cell viability. One of the bioengineering reasons for the experiment with the staining of 3D prints from the PA12 material was based on the need for a clear visual distinction of the samples in the bioreactor for future research, too.”

  1. What are the magnifications of Figures 1 and 2?

We added scaffold dimensions 45x30 mm about the square a=3mm and a wall thickness of 1mm. We did not add the magnification of Figure 1 because the photo was not taken with a microscope.

In: “2. Materials and Methods”

“Figure 1. (a) 3D-printout sample from PA12 (scaffold grid with the inner dimension of the square a=3mm and a wall thickness of 1mm)...”

“All research specimens had the same grid design (Figure 1), presenting primitive surface [12,13], lattice structures, and a grid about dimensions 45x30 mm and with a 1.5x1.5mm square component—composite scaffold with optimal gradient.”

We also added the magnification of Figure 2.:

“The images were then processed in the ImageJ-FIJI software (National Institutes of Health, USA) to quantify and characterize the differences between samples at magnification 200X (Figure 2).”

Sincerely,

Autors

Reviewer 2 Report

Comments and Suggestions for Authors

This manuscript reports the 3D-printing of a PA12 scaffold and its application for cell vitality and proliferation. The structure of the printed PA12 scaffold is quite normal with microparticles with similar sizes to cells. The impact to cell growth for PA-12 and dyed PA12 scaffold is very similar. The current experimental results are too premature, and are not able to prove the importance of this work.

Specific comments:

1.    More experiments are needed to prove the advantages of the current dyed PA12 scaffold.

2.    Experimental details are needed, e.g. cell culture plate, concentrations, etc.

3.    Some grammar mistakes, please check carefully.

Comments on the Quality of English Language

Some grammar mistakes, please check carefully.

Author Response

Dear Reviewer,

We greatly appreciate your devoted time to our manuscript. Thank you for your valuable comments on the research. We edited the study design to represent our research better and tried to add more details to make the presentation of the results more transparent.

  1. More experiments are needed to prove the advantages of the current dyed PA12 scaffold.

The primary goal of the experiment was to determine whether the stained material in the MorPA device would affect cell viability. One of the bioengineering reasons for the experiment with the staining of 3D prints from the PA12 material was based on the need for a clear visual distinction of the samples in the bioreactor for future research, too. These products are unsuitable for human use and were used only for experimental purposes. They are safe from the point of view of in vitro bioengineering applications.

During an experiment, clear and dyed PA12 3D-printed scaffolds were incubated in cell cultivation media containing essential cell growth components. It was observed that the clear PA12 granules did not change, whereas the dyed PA12 material caused a slight increase in the absorption spectrum. An MTT viability assay was conducted, which showed that metabolic activity or cell survival did not significantly reduce in the presence of clear or dyed PA12 printed parts, even after extended incubation times. Furthermore, fluorescence microscopy revealed that prolonged incubation with clear or dyed PA12 printed parts did not affect mitochondrial fitness and cell survival.

In: “3. Results”

The results of the fluorescence microscopy indicate that there was no significant difference in terms of cellular morphology, mitochondrial potential, and oxidative stress between the control cells and those exposed to clear or dyeing PA12 printed parts for 168 hours prolonged incubation at prolonged incubation with such parts did not have any negative impact on mitochondrial fitness or cell survival.

We supplemented our study by evaluating the tensile properties of plastic using standardized ASTM methods. We employed "dogbone" specimens measuring 14 millimeters in thickness and width to accomplish this. Our findings provide insight into the material's mechanical behavior under stress, which may interest academic and industry professionals alike.

In: “4. Discussion

During mechanical testing, both non-dyed (Group 1 (G1), n=30) and dyed (Group 2 (G2), n=30) PA12 samples were subjected to tensile stress. The study evaluated the tensile properties of plastic using standardized ASTM methods. It was used a Universal Testing Machine Inspect Table Blue (Hegewald Peschke Meß- und Prutechnic GmbH, Germany). To accomplish this, it employed "dogbone" specimens measuring 14 millimeters in thickness and width. The findings provide insight into the material's mechanical behavior under stress (tensile test), which may interest bioengineers and industry professionals. The breaking force (Fm [N]; G1: Fm= 2035N » G2: Fm= 2005N), elasticity modulus (E [MPa]; G1: E= 590Mpa ¹ G2: E= 1402Mpa), and relative elongation (e [mm]; G1: e = 25mm ¹ G2: e = 41mm) were measured.”

  1. Experimental details are needed, e.g. cell culture plate, concentrations, etc.

We added more information In: 2. Materials and Methodsabout experimental details, e.g. cell culture plate, concentrations, Cell viability assay, Confocal fluorescence microscopy, etc.

  1. Some grammar mistakes, please check carefully.

We apologize for the original version of the manuscript. We carefully checked grammar mistakes and other errors in the paper.

Sincerely,

Autors

Round 2

Reviewer 1 Report

Comments and Suggestions for Authors

the manuscript is suitable for publication.

Author Response

Dear reviewer.

Thank you so much for your positive review! We're glad you found the revised manuscript suitable for publication. We appreciate your helpful comments that improved the paper.

We're grateful for your time and insights. Your feedback was invaluable in making this a stronger paper.

Sincerely,

authors

Reviewer 2 Report

Comments and Suggestions for Authors

1. The importance for dyed PA12 scaffold can be emphasized in the Introduction, which will give informaton on the motivation for the preparation of dyed scaffolds.

2. How about the simplicity and universality to print scaffolds with different shape or patterns? Also, for the usage of dyes, what are the requirments for theses dyes? This can be added in the discussion section.  

Comments on the Quality of English Language

Minor editing needed. 

Author Response

Dear Reviewer,

we would like to express our sincere appreciation for the valuable advice you provided, which we have incorporated into the manuscript.

  1. The importance for dyed PA12 scaffold can be emphasized in the Introduction, which will give information on the motivation for the preparation of dyed scaffolds.

In: Introduction

"The motivation for this research was to modify the surface morphology and topography of the scaffold using a mordant, which could significantly alter the material's properties such as surface roughness, porosity reduction, support of cell adhesion, hydrophilicity, or higher mechanical resistance while maintaining the matrix's mechanical properties. One practical application of this idea is that smoothing treatment and reduced roughness could reduce tissue irritation [7] and protect against infection, which is also a reason why the research has focused on this area. Another potential application of using a mordant is to improve the ease of cleaning the surface, which could help neutralize unpleasant odors from medical devices." One of the bioengineering reasons for the experiment with the staining of 3D prints from the PA12 material was based on the need for a clear visual distinction of the samples in the bioreactor and for future research.

  1. How about the simplicity and universality to print scaffolds with different shape or patterns? Also, for the usage of dyes, what are the requirements for theses dyes? This can be added in the discussion section.

In: Discussion

"Dyes for scaffolds should be biocompatible [70], non-toxic [25], adhere to the scaffold material, be resistant to fading from light exposure, achieve the desired hue, and have minimal environmental impact [25, 46]. They should interact with scaffold material and other chemicals used in the staining process. [1]  They should have sufficient mechanical strength and structure to allow cells to settle and proliferate naturally, control their degradation and dissolution to avoid toxicity and be harmless to cells. The degree of biocompatibility is influenced by the chemical composition, structure, surface properties, and degradability of the material used.[68, 69]"

...

"In general, printing scaffolds with different morphology is a simple and universal approach used in tissue engineering. By varying the shapes and pattern of the scaffolds, can be mimic the complexity of native tissues, which promotes better cell attachment, proliferation, and differentiation. [73] It can be use various shapes of scaffolds, including simple geometric shapes like cubes and cylinders, and also more complex shapes like meshes, grooved structures, or fractal patterns. [73] It can also utilize various patterns on scaffolds, such as horizontal and vertical stripes or grooves, circular or elliptical holes, and more complex patterns like checkerboards, hexagons, or fractals. These patterns can have an impact on cell behavior and their interaction with the scaffold. [73, 63]

Several scaffold characteristics impact cell morphology, proliferation, spreading rates, and differentiation, including architecture, fiber diameter, and alignment, [71] pore size and shape, microgroove patterns, and surface roughness. [68] In general, scaffolds with high porosity and large pores tend to promote increased proliferation, differentiation, and gene expression. Pores larger than a cell can cause cells to align along a single filament, while pores smaller than a cell can lead to cell bridging across filaments, resulting in slower movement and longer migration distances. Cell behavior is dependent on the type of cell and scaffold material. [70] Changes to fiber alignment or morphology can achieve different cellular responses. However, further research is needed to confirm correlations between cell shape and fiber orientation. [68, 71]"

Thank you for your time.

Sincerely,

authors